# TMS-Induced Central Motor Conduction Time at the Non-Infarcted Hemisphere Is Associated with Spontaneous Motor Recovery of the Paretic Upper Limb after Severe Stroke

**DOI:** 10.3390/brainsci11050648

**Published:** 2021-05-15

**Authors:** Maurits H. J. Hoonhorst, Rinske H. M. Nijland, Cornelis H. Emmelot, Boudewijn J. Kollen, Gert Kwakkel

**Affiliations:** 1Rehabilitation Center Vogellanden, 8013 XZ Zwolle, The Netherlands; m.hoonhorst@vogellanden.nl; 2Amsterdam Rehabilitation Research Center|Reade, 1054 HW Amsterdam, The Netherlands; r.nijland@reade.nl; 3Department of Rehabilitation Medicine, Isala, 8025 AB Zwolle, The Netherlands; c.h.emmelot@home.nl; 4Department of General Practice and Elderly Care Medicine, University of Groningen, University Medical Center Groningen, 9712 CP Groningen, The Netherlands; b.kollen@home.nl; 5Amsterdam University Medical Center, Department of Rehabilitation Medicine, Amsterdam Movement Sciences, 1081 BT Amsterdam, The Netherlands; 6Amsterdam Neurosciences, Amsterdam University Medical Centre, 1081 HV Amsterdam, The Netherlands; 7Department of Physical Therapy and Human Movement Sciences, Feinberg School of Medicine, Northwestern University of Chicago, Evanston, IL 60208, USA

**Keywords:** stroke, recovery, upper limb, prognosis, transcranial magnetic stimulation

## Abstract

Background: Stroke affects the neuronal networks of the non-infarcted hemisphere. The central motor conduction time (CMCT) induced by transcranial magnetic stimulation (TMS) could be used to determine the conduction time of the corticospinal tract of the non-infarcted hemisphere after a stroke. Objectives: Our primary aim was to demonstrate the existence of prolonged CMCT in the non-infarcted hemisphere, measured within the first 48 h when compared to normative data, and secondly, if the severity of motor impairment of the affected upper limb was significantly associated with prolonged CMCTs in the non-infarcted hemisphere when measured within the first 2 weeks post stroke. Methods: CMCT in the non-infarcted hemisphere was measured in 50 patients within 48 h and at 11 days after a first-ever ischemic stroke. Patients lacking significant spontaneous motor recovery, so-called non-recoverers, were defined as those who started below 18 points on the FM-UE and showed less than 6 points (10%) improvement within 6 months. Results: CMCT in the non-infarcted hemisphere was prolonged in 30/50 (60%) patients within 48 h and still in 24/49 (49%) patients at 11 days. Sustained prolonged CMCT in the non-infarcted hemisphere was significantly more frequent in non-recoverers following FM-UE. Conclusions: The current study suggests that CMCT in the non-infarcted hemisphere is significantly prolonged in 60% of severely affected, ischemic stroke patients when measured within the first 48 h post stroke. The likelihood of CMCT is significantly higher in non-recoverers when compared to those that show spontaneous motor recovery early post stroke.

## 1. Introduction

There is growing evidence that stroke in one of the hemispheres also affects the so-called ‘non-lesioned hemisphere’ [1,2,3]. Recent studies indicate that these affected neuronal networks depend on the severity of post-stroke motor impairment [2,3], which may differ between primary and secondary motor networks in the non-infarcted hemisphere [2]. Support for these assumptions has been found in several serial kinematic studies investigating the quality of movement (QoM) of the less-affected limb [2,4,5]. For example, the reaching performance of the less-affected limb in patients with a stroke has been shown to be significantly slower and involving less smooth movements than that in healthy subjects, and stroke severity had a significant impact on this discoordination [4,5]. Several explanations have been proposed in the literature for the reduced QoM of the less-affected limb. Anatomically, it is known that about 10 to 15% of all corticospinal descending pathways are uncrossed or double-crossed and innervate the ipsilesional less-affected limb [6,7,8,9,10]. An alternative explanation may be that the reduced QoM is caused by the affected reticulo-, tecto- and possibly rubrospinal pathways [11,12], since these multisynaptic pathways mainly project bilaterally to the trunk and fore-arm and less to the hand muscles [13,14,15]. In support of this latter assumption, the study by Benecke and colleagues found that patients with hemispherectomy showed ipsilateral responses to transcranial magnetic stimulation (TMS) of the non-infarcted cortex suggesting activation of cortico-reticulospinal pathways [16]. In addition, the reduced QoM of the less-affected limb found in these kinematic studies may also be caused by other concomitant neurological impairments of perception and planning [2]. The reduced QoM of the less-affected limb can also possibly be explained more indirectly by a transcallosal suppression of anatomically related networks in the non-infarcted hemisphere early after a stroke [17,18,19,20]. However, the time course of this transhemispheric diaschisis in the non-infarcted hemisphere, as well as its association with the severity of stroke in the infarcted hemisphere, remain unknown so far.

Interestingly, a recent serial kinematic study showed that in most patients, the synergy-dependent coordination of the less-affected limb gradually normalizes within the first 3 months post stroke [5]. It is unknown, however, if this normalization of the QoM of the less-affected limb parallels the concomitant underlying processes of spontaneous neurobiological recovery in the infarcted hemisphere. Moreover, serial kinematic studies of the affected and less-affected limbs during a reaching or pointing task are limited to those patients who can understand and are able to perform such a task early post stroke, while it is actually the most severely affected patients with low FM-UE baseline scores early after stroke who are likely to show no spontaneous motor recovery after stroke onset [21]. In the present study, we investigated if these so-called non-recoverers are the same patients who also suffer most from a reduced central motor conduction time (CMCT) in the non-infarcted hemisphere after stroke. Unfortunately, the impact of a first-ever ischemic stroke on conductivity properties such as the CMCT of the non-infarcted hemisphere is unclear, as is the association between the absence of spontaneous motor recovery of the affected limb and the CMCT in the non-infarcted hemisphere in patients with an acute hemispheric stroke. Previous prognostic studies suggest that TMS-induced motor evoked potentials (MEPs) of the non-infarcted hemisphere are unaffected after a first-ever hemispheric stroke, regardless of stroke severity [22,23,24,25]. However, it remains unclear if the CMCT in the non-infarcted hemisphere is significantly prolonged and associated with a lack of spontaneous motor recovery of the affected limb early post stroke.

Therefore, our primary aim was to demonstrate the existence of prolonged CMCT in the non-infarcted hemisphere, measured within the first 48 h, when compared to normative data and secondly, if the severity of motor impairment of the affected upper limb is significantly associated with CMCT at the non-infarcted hemisphere within 48 h and 11 days post stroke.

## 2. Materials and Methods

### 2.1. Subjects

Patients with a first-ever ischemic hemispheric stroke as revealed by MRI or CT were prospectively screened for eligibility, and if eligible, recruited by the Isala neurology department (the Netherlands), between August 2004 and July 2007. An experienced neurologist (PvdB) classified their stroke severity and etiology within 24 h after stroke onset, using the Trial of Org 10172 in Acute Stroke Treatment (TOAST) criteria [26]. To participate in this study, patients had to show symptoms of unilateral paralysis or significant paresis of the affected upper limb (Medical Research Council score, 0–3). Patients were excluded if they (1) had a loss of consciousness, (2) had peripheral nerve pathology, including diabetes or neuromuscular disease, or (3) were unable to undergo rehabilitation because of severe comorbidity. They were also excluded if the following contraindications to TMS were present: cochlear implants, pregnancy, metal in the brain or skull, implanted neurostimulator, cardiac pacemaker or intracardiac lines and medication infusion devices [27,28].

All patients gave their verbal and/or written informed consent (themselves or by proxy) and all patients were treated according to the Dutch physical therapy guidelines for rehabilitation as soon as possible post stroke, including early mobilization and daily physical therapy interventions involving upper limb training, gait and mobility-related functions and activities [29]. The study protocol was approved by the local medical ethics board of the Isala (the Netherlands) (No 04.0318P).

### 2.2. Stimulation Procedure

The TMS technique and EMG recordings (Nihon Kohden Neuropack 8, Nihon Kohden, Tokyo, Japan) were performed according to the published guidelines [30,31]. The primary motor cortex (M1) was stimulated with a calibrated Magstim Dantec Maglite^®^ (Dantec Dynamics, Bristol, UK). Cortical TMS was applied through a figure-of-eight-shaped coil. Stimuli were administered over the non-infarcted hemisphere, see Appendix A. The hotspot that produced the highest MEP amplitude of the abductor digiti minimi (ADM) muscle of the less-affected upper limb was determined by moving the coil over the scalp in the hand area of M1 of the non-infarcted hemisphere, with the stimulator at the submaximal output. Intra- and inter-observer reliability of assessing the TMS-induced MEP of the non-infarcted hemisphere and total motor conduction time for the less-affected upper limb had been found to be good to excellent (0.45 < κ < 0.87) in 18 chronic stroke patients and 8 healthy volunteers [32].

A positive MEP was identified independently by two experienced laboratory examiners as the presence of at least three responses to three stimuli per site, producing an MEP amplitude of >50 µV [32]. When no MEP could be elicited at a given site, the coil was slightly moved to find a hotspot at adjoining sites. Cervical stimulation was performed with a 90 mm circular coil in order to activate motor roots at the exit foramina, centered over the C7/C8 cervical spine [31].

The ADM is one of the preferred intrinsic muscles that can be reliably examined as a target muscle for TMS analysis. The ADM was also chosen because of the availability of the largest collection of CMCT values in healthy subjects, which could be used as a normative reference [31]. All recordings were performed with the upper limb relaxed in order to obtain homogeneous data and to avoid variations in the muscle response due to different levels of pre-innervation [28,32]. CMCT (in milliseconds) was estimated by subtracting the peripheral motor conduction time (PMCT) from the shortest total latency time of MEPs [31,33]. PMCT was measured with the direct spinal root stimulation technique [27,31]. We defined prolonged CMCT relative to normative data obtained using the direct spinal root stimulation technique, by taking the mean (i.e., 7.1 ms) plus 2 times the standard deviation (SD) of the error, i.e., 1.1 ms [31]. Consequently, prolonged CMCT was defined as a prolongation exceeding 8.2 ms.

### 2.3. Measurements

Demographic and clinical characteristics of the participants and the TMS-induced MEPs and CMCT values were obtained within the first 48 h after a stroke (i.e., hyperacute phase) and at an average of 11 days post stroke (early subacute phase). Recently, we had found that the value of the presence of TMS-induced MEPs and the CMCT of the infarcted hemisphere in predicting upper limb motor function early after a stroke is mainly determined by the time of assessment [34]. The timing of the measurement at 11 days was chosen for practical reasons, as many patients were subsequently discharged from the department of neurology. Baseline characteristics included age, gender, left/right hemispheric stroke, etiology and stroke severity (TOAST classification) [26], motor impairment of the affected paretic upper limb (motricity index, MI) [35], presence of voluntary finger extension using the FM-UE score [36], activities of daily living using the Barthel Index (BI) [37], presence of dysphagia according to the water-swallowing test [38], and visuospatial neglect (VSN) defined as having two or more omissions in the letter cancellation task [39]. We defined patients with a lack of spontaneous motor recovery, so-called ‘non-recoverers’, in the present study as those who started at 18 points or lower on the FM-UE within the first 2 days after stroke [21,40,41] and failed to show spontaneous upper limb motor improvement beyond the smallest detectable difference (SDD) of 6 points (10%) on the FM-UE [42,43] during the first 6 months post stroke. Otherwise, subjects were classified as ‘recoverers’ in the context of spontaneous neurological recovery.

### 2.4. Data Analysis

All data were summarized using descriptive statistics. Categorical variables were described using frequencies and percentages. Continuous variables were described using means, SDs, and ranges, except for skewed variables, which are presented as medians (interquartile ranges). CMCT scores were compared between 48 h and 11 days using the Paired Student *t*-test. For processing the analysis regarding the first aim of the study, differences between recoverers and non-recoverers in CMCT scores at 48 h, categorized into normal or prolonged based on the available normative data [31], were compared using the Fischer exact test. The Mann–Whitney U test was used to describe differences between recoverers and non-recoverers in FMA motor scores at 48 h and again at 11 days. All statistical tests were conducted with IBM SPSS Statistics for Windows, version 25.0 (IBM Corp., Armonk, NY, USA). In addition, differences in FM-UE scores between recoverers and non-recoverers were assessed by plotting individual time series. For the second aim of the study, multilevel analysis was used to assess the difference between recoverers and non-recoverers regarding the longitudinal CMCT response data at 48 h and 11 days after a stroke. The iterative generalized least squares algorithm was used to estimate the regression coefficients. Patient ID was defined as level *j* and time of measurement as level *i*. A fixed slope and random intercept were selected. The Wald test was used to obtain a *p*-value for each regression coefficient. Outcome scores in the linear multilevel analysis were plotted to check for compliance with model assumptions. In multilevel logistic analysis, the outcome variable presents the logit of the probability (i.e., natural log of the odds) of prolonged longitudinal CMCT scores. Regression coefficients were subsequently transformed into odds ratios by taking the EXP [regression coefficient]. Tests were conducted using MLwiN version 3.04 (University of Bristol., Bristol, UK). A two-tailed significance level of *p* < 0.05 was used for all tests.

## 3. Results

Figure 1 shows the flowchart of screened and recruited stroke patients admitted to our hospital. After informed consent had been obtained, 51 patients with first-ever ischemic hemispheric stroke enrolled in the present prospective observational study. Fifty of them were eligible for analysis, as one patient was censored due to progressive stroke. One patient was lost to follow-up after the first assessment, but the available data of their first assessment was included in the analysis.

The main demographic and clinical characteristics are presented in Table 1. As shown in this Table, we included 29 (58%) men and 21 (42%) women, with a mean age of 70.3 years. The average age in the subgroup of recoverers (*N* = 33) was 69.7 years compared to 71.4 years for those who failed to show significant recovery beyond the 6 points improvement on the FM-UE (*N* = 17). As shown in Table 2 and illustrated in Figure 2, the recoverers improved their median FM-UE total score from 35 points within 48 h to 61 points at 6 months. The median FM-UE score of non-recoverers was 3 points within 48 h and showed a median improvement of 2 points at 6 months after stroke. Eighteen out of the 50 (36%) patients showed some voluntary finger extension within 48 h, whereas 32 out of 50 (64%) patients did not. All non-recoverers failed to show voluntary finger extension within the first 48 h after stroke. In addition, 14 out of the 17 (87.5%) non-recoverers failed to show active finger extension at 11 days, compared with 9 out of 33 (27%) recoverers. Finally, recoverers were significantly less likely to have visuospatial neglect (*p* < 0.001), had shorter hospital stays (*p* < 0.001) and significantly higher BI scores (*p* < 0.001) compared to the non-recoverers.

Within 48 h post stroke, 30 out of 50 (60%) patients showed prolonged CMCT in the non-infarcted hemisphere. After 11 days, the number of patients with prolonged CMCT was reduced to 24 out of 49 (49%). Mean CMCT (±SD) of all 50 patients within 48 h post stroke was 10.3 ms (±4.35), which was statistically significantly longer than the CMCT at 11 days after a stroke: 9.44 ms (±3.99); *p* = 0.044. At 48 h after stroke, significantly more patients with spontaneous motor recovery after 6 months showed a normal CMCT in the non-infarcted hemisphere when compared to the non-recoverers (*p* = 0.032).

Multilevel analysis of CMCT (as a continuous variable) between recoverers and non-recoverers, comparing two measurements at 48 h and 11 days after stroke, revealed no statistically significant differences between these subgroups (lower CI = −4.146; upper CI = 0.228; *p* = 0.079). Multilevel logistic analysis yielded statistically significant differences between the recoverers and non-recoverers as regards the longitudinal association with a normal and prolonged CMCT response, respectively, from 48 h to 11 days post stroke (lower CI = 0.086; upper CI = 0.763; *p* = 0.015), with an odds ratio of 0.256 (95%CI: 0.086–0.763).

## 4. Discussion

In the present study, we showed for the first time that in the majority (60%) of patients with a severe upper limb impairment, CMCTs in the non-infarcted hemisphere are significantly prolonged when measured within the first 48 h post stroke. In the hyperacute phase after stroke, prolonged CMCT is associated with the severity of motor impairment of the affected upper limb. More importantly, about half of the patients with prolonged CMCTs in the non-infarcted hemisphere within the first 48 h after stroke will have persistent, prolonged conduction times within the first 11 days. In patients with a severe, first-ever, ischemic hemispheric stroke, the likelihood that CMCTs remain prolonged during this 11 day time window and do not normalize to conduction times seen in healthy age-matched subjects is significantly associated with the absence of significant spontaneous motor recovery (i.e., non-recovery) of the affected upper limb according to FM-UE scores. The odds of prolonged CMCT in the first 11 days are 0.256 for recoverers, compared with the odds of a prolonged CMCT for non-recoverers. This finding suggests that the odds of prolonged CMCT in the subgroup of recoverers are, on average, 74.4% lower than in the subgroup of non-recoverers. Finally, our findings confirm the results of previous work by Byblow and colleagues [44], showing that the integrity of the CST for the less-affected limb remains functionally intact irrespective of stroke severity. Our finding of a significantly prolonged CMCT is also in agreement with recent kinematic studies, in which significant changes in, for example, speed and intralimb coordination were found for the less-affected limb in the first weeks after stroke [4,5].

Question of how to explain the above findings regarding prolonged CMCTs in the non-infarcted hemisphere and their association with the proportional level of spontaneous motor recovery remains unanswered in the present study. As a first hypothesis, one may assume that anatomically related networks in the non-infarcted hemisphere of severely affected stroke patients are temporarily suppressed in the acute phase, probably by transcallosal diaschisis [17,19,45,46]. Although poorly understood, this disturbed function in anatomically associated areas is believed to be caused by inflammation and oxidative stress due to an upregulation of cytokines and increased activity of macrophages and glial cells in areas anatomically associated with the infarcted area [47,48,49]. In line with this hypothesis, recent longitudinal diffusion tension imaging [50] and fMRI-resting state studies investigating functional connectivity [51] also found evidence for transhemispheric cortical and white matter changes in anatomically related areas very early post stroke. For example, Visser and colleagues showed significant changes in white matter integrity in the ipsi- as well as contralesional brain areas, such as the primary motor, pre-motor and visual cortices, during the first months after a stroke [50]. Theoretically, an alternative explanation may be that the decline of CMCT in the non-infarcted hemisphere between 48 h and 11 days reflects the underlying recovery of the anatomically uncrossed or double-crossed CST and the reticulo-, tecto- and possibly rubrospinal descending pathways. However, the latter explanation is unlikely, as the ipsilateral CST mainly projects to the trunk and upper arm [11,12,52], whereas the recovery of multisynaptic reticulospinal pathways is too slow for CMCT speeds below 8.2 ms [16,53].

Our observational study had some limitations. First, the sample of stroke subjects (*N* = 50) is too small to find robust findings, even though the current study is one of the largest prospective cohorts in this field to start within 48 h after a stroke. Second, we did not combine the present findings with neuroimaging techniques such as CT or MRI angiography or CT/MRI perfusion imaging, which would have allowed us to identify the severity of irreversible brain damage in the hyperacute phase post stroke [54], or to detect the existence of mass effects by central or cingulate shifts affecting the hemodynamics in the non-infarcted hemisphere early post stroke. However, this latter mechanism is unlikely, in our opinion, since the evolution of mass effects by vasogenic oedema requires several days after hemispheric stroke [55] and is accompanied by reduced consciousness of the patient. Third, the cut-off score for prolonged CMCT was obtained from normative laboratory values [31]. Although this is a rather conservative CMCT criterion, it can still be criticized as being susceptible to variation and bias. The used normative data of healthy subjects were specifically obtained with the ADM as target muscle [31]. Age could potentially be a confounder in the association between CMCT and observed improvement. However, the significant association between CMCT of the non-infarcted hemisphere and improvements in FM-UE scores was not significantly influenced in our data after (partial) correcting for age, acknowledging that age was statistically different between recoverers and non-recoverers at baseline in our sample.

Finally, we did not investigate the impact of the type and intensity of rehabilitation as possible confounders of motor recovery in the first 6 months post stroke. All patients received usual care according to the current Dutch guidelines for stroke rehabilitation [29]. However, there is currently no evidence that exercise therapy can significantly influence the time course of spontaneous motor recovery early post stroke [44,56]. In the present study, we provide additional evidence that the CMCT in the non-infarcted hemisphere is prolonged in a large proportion of patients when measured in the hyperacute phase after a severe stroke. In addition, we found that a prolonged CMCT in the non-infarcted hemisphere persisted during the first 11 days in those who showed no spontaneous motor recovery in the most affected upper limb post stroke.

## 5. Conclusions

The current study suggests that CMCT in the non-infarcted hemisphere is significantly prolonged in 60% of severely affected, ischemic stroke patients when measured within the first 48 h post stroke. The likelihood of prolonged CMCT is significantly higher in those patients that show no spontaneous motor recovery (i.e., ‘non-recoverers’) when compared to those that show spontaneous motor recovery (i.e., ‘recoverers’) early after a first-ever ischemic hemispheric stroke. 

## Figures and Tables

**Figure 1 brainsci-11-00648-f001:**
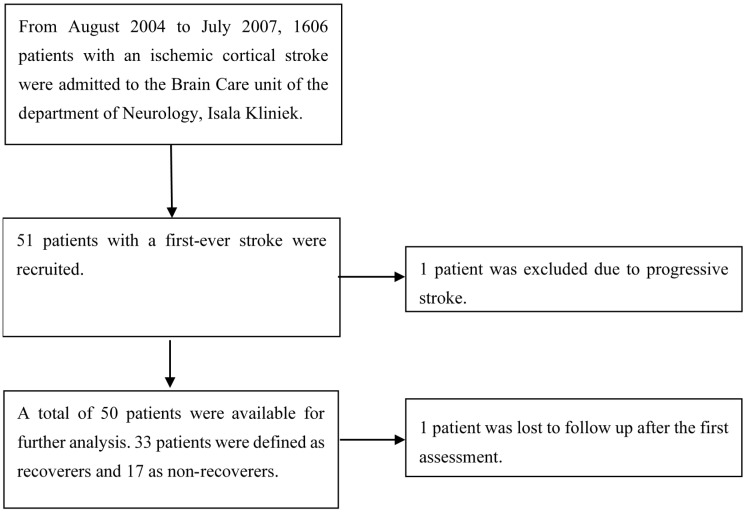
Patient exclusion flowchart.

**Figure 2 brainsci-11-00648-f002:**
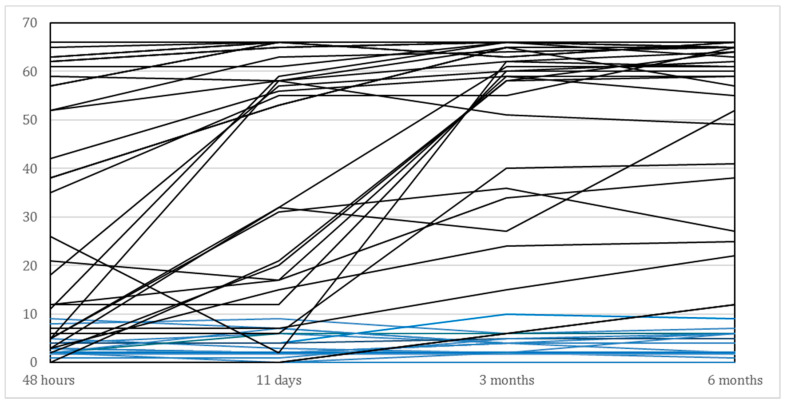
Time course of individual FM-UE scores for the affected upper limb of all patients (*N* = 50), between baseline (i.e., within 48 h after stroke) and 6 months post stroke. Blue lines represent the non-recoverers (*N* = 17) and black lines those who showed spontaneous neurological recovery (*N* = 33).

**Table 1 brainsci-11-00648-t001:** Patient characteristics of the total group (*N* = 50, second column), and subgroups (right-hand columns) in relation to spontaneous neurological upper limb motor recovery early after stroke.

	Total Group	Recoverers (*N* = 33)	Non-Recoverers (*N* = 17)	*p*
Gender, F/M	29/21	17/16	12/5	0.571
Age, mean (SD), yr	70.3 (12.3)	69.7 (12.8)	71.4 (11.4)	<0.0001 *
Hemisphere of stroke, L/R	25/25	16/17	9/8	0.170
Length of hospital stay, mean (range), d	4.9 (6–38)	13.1 (6–25)	15.7 (6–38)	<0.0001 *
Type of Stroke (TOAST): LVD/SVD/undetermined	34/14/2	22/10/1	12/4/1	0.804
Dysphagia at 48 h, yes/no	33/17	20/13	13/4	1.00
VSN at 48 h, yes/no	12/38	7/26	5/12	<0.0001 *
Barthel index score (0–20) at 48 h, median (IQR)	5.0 (5.5)	7 (5)	2 (4)	<0.0001 *

The subgroups are stroke patients with (‘recoverers’) or without (‘non-recoverers’) upper limb motor recovery at 6 months post stroke. Non-recoverers were defined as those patients who started at 18 points or lower on the Fugl-Meyer upper extremity scores for the affected limb within the first 48 h, and showed improvements of 6 points or less within the first 6 months post stroke. Abbreviations: F, female; M, male; L/R, left or right affected hemisphere; TOAST, Trial of Org 10172 in Acute Stroke Treatment; LVD, large vessel disease; SVD, small vessel disease; VSN, visuospatial neglect; IQR, interquartile range. * *p* < 0.05.

**Table 2 brainsci-11-00648-t002:** Transcranial Magnetic Stimulation and Fugl-Meyer upper extremity score characteristics for the total group (*N* = 50, second column), and subgroups (third and fourth columns) in relation to spontaneous neurological upper limb motor recovery after stroke.

	Total Group	Recoverers (*N* = 33)	Non-Recoverers (*N* = 17)	*p*
CMCT at 48 h, mean (SD), ms	10.27 (4.35)	9.54 (2.95)	11.7 (6.10)	<0.0001 *
CMCT at 11 days, mean (SD), ms	9.44 (3.99)	8.81 (3.17)	10.7 (5.18)	<0.0001 *
FM-UE finger extension at 48 h, yes/no	18/32	18/15	0/17	<0.0001 *
FM-UE finger extension at 11 days, yes/no	26/23 §	24/9	2/14	0.065
FM-UE total score (0–66), median (IQR) at 48 h	7.5 (43)	35 (53.5)	3 (2.5)	<0.0001 *
FM-UE total score (0–66), median (IQR) at 6 months	54 (58) §	61 (12)	5 (3.8)	<0.0001 *

The third and fourth columns show subgroup comparisons between stroke patients with (‘recoverers’) or without (‘non-recoverers’) upper limb motor recovery at 6 months post stroke. Non-recoverers were defined as those patients who started at 18 points or lower on the FM-UE within the first 48 h and showed improvements of 6 points or less within the first 6 months post stroke. Abbreviations: CMCT, central motor conduction time of the non-infarcted hemisphere; Prolonged CMCT was defined as a latency > 8.2 ms. FM-UE, Fugl-Meyer upper limb motor score. IQR, interquartile range.* *p* < 0.05. § 1 lost to follow-up.

## Data Availability

The data presented in this study are available on request from the corresponding author. The data are not publicly available due to practical, ethical and privacy reasons.

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
