# Peer review of "TMS-Induced Central Motor Conduction Time at the Non-Infarcted Hemisphere Is Associated with Spontaneous Motor Recovery of the Paretic Upper Limb after Severe Stroke"

_brainsci, 2021, doi:10.3390/brainsci11050648_

Round 1

Reviewer 1 Report

Thank you for the opportunity to review this manuscript. In general, it is well written and interesting. I have a couple of key points that I would like the authors to consider and respond to.

  1. My biggest concern is regarding normative data which was used to compare CMCT in the stroke participants. The normative data comes from reference 33 and I have not been able to obtain this full text article. I have asked to obtain it from multiple sources and it is proving very difficult to obtain, therefore making it difficult to complete an appropriate assessment of the accuracy of this work. However, I note from the abstract of this work, the reported reference values of mean 7.1ms and SD 1.1ms refer to a CMCT time assessed with facilitation. I am unsure what that facilitation is (as I cannot access the article), but am curious if it was pre-activation of the motor pathway (ie arm in an active condition). This would appear plausible as the CMCT time is shorter in facilitation than at rest in this reference paper. If this is the case, then the reference values are not appropriate for your study given you did not replicate this procedure.
  2. Along similar lines - I cannot determine the age of participants in your reference values (as I cannot access the study). Were these participants of similar age to the stroke participants in your study. Noting of course that age does have an effect on CMCT (see Cantone et al 2019, Front Hum Neurosci).
  3. Linked to the above point - Age is statistically different between your recoverers and non-recoverers. Why was age not controlled for when comparing CMCT?
  4. lines 248-260 presents a lot of repetition of data in tables. Suggest removing the bulk of it.
  5. Please review lines 355-360. Appears to be a language issue here.
  6. I am not clear (and partly concerned) by the use of the reference dataset for CMCT times. Why did the authors simply not correlate CMCT with change in FM. I suggest this is a much more appropriate analysis for this dataset.

Author Response

Reviewer

Thank you for the opportunity to review this manuscript. In general, it is well written and interesting. I have a couple of key points that I would like the authors to consider and respond to.

  1. Reviewer

My biggest concern is regarding normative data which was used to compare CMCT in the stroke participants. The normative data comes from reference 33 and I have not been able to obtain this full text article. I have asked to obtain it from multiple sources and it is proving very difficult to obtain, therefore making it difficult to complete an appropriate assessment of the accuracy of this work. However, I note from the abstract of this work, the reported reference values of mean 7.1ms and SD 1.1ms refer to a CMCT time assessed with facilitation. I am unsure what that facilitation is (as I cannot access the article), but am curious if it was pre-activation of the motor pathway (ie arm in an active condition). This would appear plausible as the CMCT time is shorter in facilitation than at rest in this reference paper. If this is the case, then the reference values are not appropriate for your study given you did not replicate this procedure.

Answer:

We like to thank this reviewer for his/her valuable comments.

We mainly based our normative reference data on the extensive publication of Groppa and colleagues 1 who present a comprehensive overview of the literature with normative CMCT data for the upper limb. Groppa and colleagues based their normative values (for different target muscles) on several references. We first added the study of Pajaron et al 2 to our reference list because they presented normative CCT data, both with and without facilitation, also based on suboptimal TMS intensity as we used in our stimulation protocol. With CMCT values without facilitation, ranging from about 7 to 10 milliseconds. Unfortunately, the original paper of Pajaron is written in Spanish, therefore, we decided to remove this manuscript from the reference list.

Also, when taking slower CMCT values as a reference such as the upper 95%CI, which is 10 msec, we were still able to distinguish our stroke sample from the reference value. We will address above reviewers concern with respect to our used (not age-matched) reference values for CMCT as a limitation in the discussion. We therefore added the following text (also with regard to the topics of points 2 and 3 as stated below,) at lines 382-388 in the discussion section:

"The used normative data of healthy subjects was obtained with the ADM as target muscle 31. Age could potential be a confounder in the association between CMCT and observed improvement. However, the significant association between CMCT on the less affected side and improvements in FM-UE scores was not significantly influenced in our data after (partial) correcting for age, acknowledging that age was statistically different between recoverers and non-recoverers at baseline in our sample."

  1. Reviewer

Along similar lines - I cannot determine the age of participants in your reference values (as I cannot access the study). Were these participants of similar age to the stroke participants in your study. Noting of course that age does have an effect on CMCT (see Cantone et al 2019, Front Hum Neurosci).

Answer:

We understand the concern of the possible influence of age on CMCT and the issue of lacking age-matched data from used reference values.          Indeed, Cantone and colleagues found a correlation between age and CMCT 3.

However, it should be noted that this influence of age on CMCT is debatable 1,4,5 . Following the report of the IFCN committee, age has a negligible effect on CMCT measures, provided that the TMS stimulus intensity is sufficiently high 4.    

Related to this question, we subsequently analyzed if age could be of influence on the correlation of FM-UE change scores from 48 hours till 6 months with CMCT values at 48 hours with and without correction for age. However, the bivariate association between CMCT and FM-UE change was not significantly influenced by age, calculated with a partial correlation coefficient, corrected for age.

In line with previous reviewers comment, we, therefore, added the following text at lines 380-39 in the discussion section:

"The used normative data of healthy subjects was obtained with the ADM as target muscle 31. Age could potential be a confounder in the association between CMCT and observed improvement. However, the significant association between CMCT on the less affected side and improvements in FM-UE scores was not significantly influenced in our data after (partial) correcting for age, acknowledging that age was statistically different between recoverers and non-recoverers at baseline in our sample."

  1. Reviewer

Linked to the above point - Age is statistically different between your recoverers and non-recoverers. Why was age not controlled for when comparing CMCT?

Answer:

We analyzed if age could be of influence on the correlation of CMCT at day 11 after stroke between the subgroups of recoverers and non-recoverers. We choose to use CMCT data at 11 days for this analysis, because CMCT values at 24 hours change significantly in the first week and the proportion between prolonged and normal CMCT values at days 11 seems not alter that much. The bivariate association between CMCT at 11 days and subgroups was not significantly influenced if corrected for age.

We added text at lines 382-388 in the discussion section:

"The used normative data of healthy subjects was obtained with the ADM as target muscle 31. Age could potential be a confounder in the association between CMCT and observed improvement. However, the significant association between CMCT on the less affected side and improvements in FM-UE scores was not significantly influenced in our data after (partial) correcting for age, acknowledging that age was statistically different between recoverers and non-recoverers at baseline in our sample."

  1. Reviewer

lines 248-262 presents a lot of repetition of data in tables. Suggest removing the bulk of it. 

ANSWER:

We agree with the reviewer and made this text leaner by referring to Tables 1 and 2 and deleting redundant estimates such as IQR and SDs in the text. We also added text about finger extension at lines 259-261 which is clinically relevant for the readers, in our opinion:

" In addition, 14 out of the 17 (87.5%) non-recoverers failed to show active finger extension at 11 days, compared with 9 out of 33 (27%) recoverers."

The results are rewritten in the following way at lines 252-266:

"The main demographic and clinical characteristics are presented in Table 1. As shown, we included 29 (58%) men and 21 (42%) women, with a mean age of 70.3 years. The average age in the subgroup of recoverers (N= 33) was 69.7 years compared to 71.4 years for the non-recovers who failed to show significant recovery beyond the 6 points improvement on the FM-UE (N=17).  As shown in Table 2 and illustrated in Figure 2, the recoverers improved their median FM-UE total score from 35 points within 48 hours to 61 points at 6 months. The median FM-UE score of non-recoverers was 3 points within 48 hours. They showed a median improvement of 2 points at 6 months after stroke. Eighteen out of the 50 (36%) patients showed some voluntary finger extension within 48 hours, whereas 32 out of 50 (64%) patients did not. All non-recoverers failed to show voluntary finger extension within the first 48 hours after stroke. In addition, 14 out of the 17 (87.5%) non-recoverers failed to show active finger extension at 11 days, compared with 9 out of 33 (27%) recoverers. Finally, recoverers were significantly less likely to have visuospatial neglect (P<0.001), had shorter hospital stays (P<0.001) and significantly higher BI scores (P<0.001) compared to the non-recoverers."

  1. Reviewer

Please review lines 355-360. Appears to be a language issue here.

Answer:

At lines 356 and 357 the two words "infarcted hemisphere", were deleted.

  1. Reviewer

I am not clear (and partly concerned) by the use of the reference dataset for CMCT times. Why did the authors simply not correlate CMCT with change in FM. I suggest this is a much more appropriate analysis for this dataset.

Answer:

We agree with this reviewer, and added the non-significant influence of age as a potential

confounder on the association between CMCT and observed improvements on FM-UE scores on

the most affected side as a limitation in the discussion, lines 383-388 (See also answer 3):

Age could potential be a confounder in the association between CMCT and observed improvement.

However, the significant association between CMCT on the less affected side and improvements in FM

UE scores was not significantly influenced in our data after (partial) correcting for age, acknowledging

that age was statistically different between recoverers and non-recoverers at baseline in our sample."

1)         S. Groppa, A. Oliviero, A. Eisen, A. Quartarone, L.G. Cohen, V. Mall, A. Kaelin-Lang, T. Mima, S.      Rossi, G.W. Thickbroom, P.M. Rossini, U. Ziemann, J. Valls-Solé, H.R. Siebner. A practical guide           to diagnostic transcranial magnetic stimulation: Report of an IFCN committee.  Clin Neurophys         2012;123 (5): 858-882.

2)         E. Pajarón et al., “Reference values for the central motor conduction time and silent period   obtained by the transcranial magnetic stimulation,” Rev Neurol , vol. 28, no. 3, pp. 227–232, 1999.

3)         Cantone M, Lanza G, Vinciguerra L, Puglisi V, Ricceri R, Fisicaro F, Vagli C, Bella R, Ferri R,           Pennisi G, Di Lazzaro V, Pennisi M. Age, Height, and Sex on Motor Evoked Potentials:        Translational Data From a Large Italian Cohort in a Clinical Environment. Front Hum Neurosci.         2019;13:185. doi: 10.3389/fnhum.2019.00185.

4)         Wassermann EM. Variation in the response to transcranial magnetic brain stimulation in the

            general population. Clin Neurophysiol. 2002 Jul;113(7):1165-1171. doi: 10.1016/s1388-       2457(02)00144-x.

5)         Rossini PM, Berardelli A, Deuschl G, Hallett M, Maertens de Noordhout AM, Paulus W, Pauri F.

             Applications of magnetic cortical stimulation. The International Federation of Clinical

             Neurophysiology. Electroencephalogr Clin Neurophysiol Suppl. 1999;52:171-85. PMID:

             10590986.

6)         Carle AC. Fitting multilevel models in complex survey data with design weights:

             Recommendations. BMC Med Res Methodol. 2009;9:49. doi: 10.1186/1471-2288-9-49.

             PMID: 19602263; PMCID: PMC2717116.

Reviewer 2 Report

This study assessed the relationship between contralesional central motor conduction time (CMCT) and motor impairment and recovery. The authors were interested in whether delays in CMCT in the first 48 hours after stroke related to motor impairment and whether if this prolongation was sustained in the first two weeks following stroke, there was a relationship with spontaneous motor recovery measured 6 months post-stroke. Individuals were divided into ‘recoverers’ and ‘non-recoverers’ for further analyses. The main results from the study suggest that CMCT was delayed at both timepoints post-stroke (in 60% at 48 hours, and 49% at 11 days), with this pattern being more frequent in individuals determined to be ‘non-recoverers’.

Introduction
The rational for the use of CMCT as opposed to another measure of neurophysiology in the contralesional hemisphere should be emphasized more in the introduction. The authors do a good job outlining the behavioural deficits that can be seen in the less affected limb, and this would plausibly be underpinned by altered physiology, but why the CMCT is the most appropriate measure to test this is not obvious. 

The aims are stated in such a way that there is no question that the CMCT will be delayed in individuals post-stroke. Has this been shown extensively in previous literature or should one aim be to determine contalesional CMCT is delayed compared to normative data, and then to explore the relationship between this measure and impairment/recovery? In the stats it seems that the authors do compare their data with normative data, so I would suggest changing the phrasing to match this. That could then be followed with the hypothesis that the CMCT would be prolonged, and the expected relationship with impairment and recovery could also be added as a hypothesis. 

Methods
Please provide more information on the methods used to determine CMCT. For instance, what intensity was used for stimulation? It is stated that positive MEP was defined as presence of at least three responses to three stimuli. To determine the CMCT latency, was an average of these responses taken or was the earliest response taken? 

Based on the authors’ description in the text and the data depicted in Figure 2, it seems that a large number of individuals in the ‘recoverers’ group have high FM scores at 48 hours, but the recovery trajectory mirrors that seen in many of the ‘non-recoverers’ in that it looks like there is not a great increase in function over time. Of course, this is partially due to a ceiling effect in individuals that start very high as there is not the same amount of room for improvement, but I’m wondering if these individuals should be treated the same as, for instance, the individuals who truly seem to fit the definition provided for ‘recoverers’ who meaningfully improve from low initial FM scores. If the analyses were run with a third subgroup for these individuals, or if they were removed from the analysis altogether, are the results any different?

Generally in the stats section, it might be useful for the reader if the stats run were linked to the aims presented in the introduction in a more explicit manner.  

On line 180 the text reads that a fixed and random intercept was selected. I’m wondering if this should read a fixed slope and random intercept?

Results
Please provide full statistical results beyond just the p-values. 

Given that the aims are about the relationship between CMCT and impairment/recovery, a figure with this data and that speaks to this relationship should be included in the results.

Author Response

Comments and Suggestions for Authors

This study assessed the relationship between contralesional central motor conduction time (CMCT) and motor impairment and recovery. The authors were interested in whether delays in CMCT in the first 48 hours after stroke related to motor impairment and whether if this prolongation was sustained in the first two weeks following stroke, there was a relationship with spontaneous motor recovery measured 6 months post-stroke. Individuals were divided into ‘recoverers’ and ‘non-recoverers’ for further analyses. The main results from the study suggest that CMCT was delayed at both timepoints post-stroke (in 60% at 48 hours, and 49% at 11 days), with this pattern being more frequent in individuals determined to be ‘non-recoverers’.

Reviewer

  1. Introduction
    The rational for the use of CMCT as opposed to another measure of neurophysiology in the contralesional hemisphere should be emphasized more in the introduction. The authors do a good job outlining the behavioural deficits that can be seen in the less affected limb, and this would plausibly be underpinned by altered physiology, but why the CMCT is the most appropriate measure to test this is not obvious. 

The aims are stated in such a way that there is no question that the CMCT will be delayed in individuals post-stroke. Has this been shown extensively in previous literature or should one aim be to determine contalesional CMCT is delayed compared to normative data, and then to explore the relationship between this measure and impairment/recovery? In the stats it seems that the authors do compare their data with normative data, so I would suggest changing the phrasing to match this. That could then be followed with the hypothesis that the CMCT would be prolonged, and the expected relationship with impairment and recovery could also be added as a hypothesis. 

Answer:

We agree with the reviewer that the first step in this rationale would be to demonstrate if CMCT is actually prolonged when compared to normative data. Therefore, we changed the text in the abstract (lines 23-24) and the introduction section (lines 89 and 91) in:

"Therefore, our primary aim was to demonstrate the existence of prolonged CMCT in the non-infarcted hemisphere, measured within the first 48 hours, when compared to normative data and secondly, if severity of motor impairment of the affected upper limb is associated with CMCT at the non-infarcted hemisphere in the first 2 weeks post stroke. "

  1. Reviewer

Methods
Please provide more information on the methods used to determine CMCT. For instance, what intensity was used for stimulation? It is stated that positive MEP was defined as presence of at least three responses to three stimuli. To determine the CMCT latency, was an average of these responses taken or was the earliest response taken? 

Answer:

We agree with the reviewer the procedure for measuring CMCT need more details. Therefore, we added how CMCT was estimated and based on the shortest latency. Accordingly, we changed the text a line 140:

"CMCT (in milliseconds) was estimated by subtracting the peripheral motor conduction time (PMCT) from the shortest latency time of MEPs. "

  1. Reviewer

Based on the authors’ description in the text and the data depicted in Figure 2, it seems that a large number of individuals in the ‘recoverers’ group have high FM scores at 48 hours, but the recovery trajectory mirrors that seen in many of the ‘non-recoverers’ in that it looks like there is not a great increase in function over time. Of course, this is partially due to a ceiling effect in individuals that start very high as there is not the same amount of room for improvement, but I’m wondering if these individuals should be treated the same as, for instance, the individuals who truly seem to fit the definition provided for ‘recoverers’ who meaningfully improve from low initial FM scores. If the analyses were run with a third subgroup for these individuals, or if they were removed from the analysis altogether, are the results any different?

Answer:

We agree that in the subgroup of recoverers, 6 subjects had a high baseline with initial FM-UE scores of 60 points or more. These patients suffer from ceiling effects and were not able to improve 6 points or more on the FM-UE.  Therefore, we re-analyzed if the bivariate association between CMCT at 11 days and FM-UE change scores and found that the outcome was not significantly influenced with and without including these 6 subjects that had baseline score of 60 points or higher.

  1. Reviewer

Generally in the stats section, it might be useful for the reader if the stats run were linked to the aims presented in the introduction in a more explicit manner.  

Answer:

To link the used stats better to the aims of our study, we inserted the following text:

In lines 171-172 to the text:

For processing the analysis with regard to the first aim of our study, differences between recoverers and non-recoverers in CMCT scores at 48 hours, categorized into normal or prolonged based on the available normative data [31], [33] were compared using the Fischer exact test.”

In line 178 to the text:

 “For the second aim of the study, multilevel analysis was used to assess the difference between recoverers and non-recoverers with regard to the longitudinal CMCT response data at 48 hours and 11 days after stroke. “

  1. Reviewer

On line 180 the text reads that a fixed and random intercept was selected. I’m wondering if this should read a fixed slope and random intercept?

Answer:

We do apologize for this confusion. Indeed, the text at lines 183 should be: A fixed slope and random intercept was selected.

  1. Reviewer

Results
Please provide full statistical results beyond just the p-values. 

Answer:

To provide more statistical information, we added the following besides the p-value in lines 322-323:

"lower CI= -4.146; upper CI= 0.228."

With the same purpose, we added the following text in line 326:

 "lower CI= 0.086; upper CI= 0.763."

  1. Reviewer

Given that the aims are about the relationship between CMCT and impairment/recovery, a figure with this data and that speaks to this relationship should be included in the results.

Answer:

Proportion of subjects with prolonged and normal CMCT values within 48 hours post stroke.

Two-tailed P-value following the Fisher Exact Test: .0323

1) van der Vliet R, Selles RW, Andrinopoulou ER, Nijland R, Ribbers GM, Frens MA, Meskers C, Kwakkel G. Predicting Upper Limb Motor Impairment Recovery after Stroke: A Mixture Model. Ann Neurol. 2020 Mar;87(3):383-393. doi: 10.1002/ana.25679. Epub 2020 Jan 25. PMID: 31925838; PMCID: PMC7065018.

Round 2

Reviewer 2 Report

The authors have addressed all the comments that I had in the previous round. The paper should be read to check for minor errors such as typos, or missing words. One example is that I think there is a word missing in the final sentence of the abstract. Other than that the paper is clear and well written and I thank the authors for incorporating my comments.